# Hyaluronic Acid in Synovial Fluid Prevents Neutrophil Activation in Spondyloarthritis

**DOI:** 10.3390/ijms24043066

**Published:** 2023-02-04

**Authors:** Sanne Mol, Esther W. M. Taanman-Kueter, Baltus A. van der Steen, Tom Groot Kormelink, Marleen G. H. van de Sande, Sander W. Tas, Marca H. M. Wauben, Esther C. de Jong

**Affiliations:** 1Department of Experimental Immunology, Amsterdam Institute for Infection and Immunity, Amsterdam UMC, Location AMC, 1105 AZ Amsterdam, The Netherlands; 2Department of Biomolecular Health Sciences, Faculty of Veterinary Medicine, Utrecht University, 3584 CL Utrecht, The Netherlands; 3Department of Rheumatology and Clinical Immunology, Amsterdam Rheumatology and Immunology Center, Amsterdam University Medical Centers, University of Amsterdam, 1105 AZ Amsterdam, The Netherlands

**Keywords:** neutrophils, neutrophil activation, spondyloarthritis, synovial fluid, hyaluronic acid

## Abstract

Spondyloarthritis (SpA) patients suffer from joint inflammation resulting in tissue damage, characterized by the presence of numerous neutrophils in the synovium and synovial fluid (SF). As it is yet unclear to what extent neutrophils contribute to the pathogenesis of SpA, we set out to study SF neutrophils in more detail. We analyzed the functionality of SF neutrophils of 20 SpA patients and 7 disease controls, determining ROS production and degranulation in response to various stimuli. In addition, the effect of SF on neutrophil function was determined. Surprisingly, our data show that SF neutrophils in SpA patients have an inactive phenotype, despite the presence of many neutrophil-activating stimuli such as GM-CSF and TNF in SF. This was not due to exhaustion as SF neutrophils readily responded to stimulation. Therefore, this finding suggests that one or more inhibitors of neutrophil activation may be present in SF. Indeed, when blood neutrophils from healthy donors were activated in the presence of increasing concentrations of SF from SpA patients, degranulation and ROS production were dose-dependently inhibited. This effect was independent of diagnosis, gender, age, and medication in the patients from which the SF was isolated. Treatment of SF with the enzyme hyaluronidase strongly reduced the inhibitory effect of SF on neutrophil activation, indicating that hyaluronic acid that is present in SF may be an important factor in preventing SF neutrophil activation. This finding provides novel insights into the role of soluble factors in SF regulating neutrophil function and may lead to the development of novel therapeutics targeting neutrophil activation via hyaluronic acid or associated pathways.

## 1. Introduction

Chronic inflammatory joint diseases are highly prevalent with estimates ranging from 5–20% depending on the subtypes included [1]. In addition to the most common type of arthritis, i.e., rheumatoid arthritis (RA), spondyloarthritis (SpA) is the second most prevalent inflammatory joint disease that affects roughly 0.2–1.6% of the world population [2]. SpA patients suffer from pain and stiffness in the spine and/or peripheral joints due to inflammation and structural damage. They may also exhibit inflammation of the bowel mucosa, eyes (i.e., anterior uveitis), and the skin (i.e., psoriasis) [2,3,4,5]. Current effective treatments are non-steroidal anti-inflammatory drugs (NSAIDs) aimed at inhibiting cyclooxygenase, conventional synthetic disease-modifying anti-rheumatic drugs (csDMARDs) such as methotrexate or sulfasalazine (for peripheral joints), and biological or targeted synthetic DMARDs (bDMARDs) that block pro-inflammatory cytokines such as tumor necrosis factor (TNF), or IL-17A, or intracellular signal transduction pathways such as JAK/STAT signaling [6,7,8].

In the inflamed joints of SpA patients, neutrophils are present in high numbers both in the synovial tissue and in the synovial fluid (SF) [9]. Neutrophils are the most common leukocyte subset in blood and one of the first responders of the host defense system during an infection. The main function of neutrophils is to capture and destroy invading pathogens such as bacteria and fungi. To do so, they are equipped with various effector functions such as degranulation, mediator release, reactive oxygen species (ROS) production, phagocytosis, and neutrophil extracellular trap (NET) formation [10,11]. However, neutrophils are also associated with various immune-mediated inflammatory diseases, including inflammatory bowel disease (IBD) [12,13,14]. In IBD, a disease closely related to SpA, disease severity correlates with the number of invasive neutrophils [12]. Furthermore, it was reported that the mucosa of IBD patients exhibits higher levels of neutrophil elastase compared to healthy controls, indicating the local presence of active neutrophils [15]. The abundant presence of neutrophils in psoriatic skin lesions serves as a typical histopathologic hallmark of psoriasis [16]. The neutrophil-to-lymphocyte ratio (NLR), the activity of neutrophils, and the number of NETotic cells were significantly higher in psoriasis patients compared to healthy controls [17]. In RA and SpA, the severity of the disease correlates with an increased neutrophil-to-lymphocyte ratio [13]. In addition, neutrophils present in SF of RA patients produce more reactive oxygen species (ROS), and the formation of neutrophil extracellular traps (NETosis) is increased [18]. The observation that IL-17 levels in SF of SpA patients are increased compared to levels in SF of OA patients suggests that neutrophils may be involved in Th17 activation, the main producers of IL-17 [19,20].

Although there is evidence for the presence and involvement of neutrophils in various forms of arthritis, the exact role of neutrophils in the disease process is still elusive. Therefore, we set out to study SF neutrophils derived from SpA patients in more detail. First, we characterized surface markers involved in neutrophil degranulation. Here, we demonstrated that, surprisingly, neutrophils present in SF of SpA patients are neither activated nor exhausted, as they can respond in vitro to stimuli by degranulation and ROS production. We show that soluble factors present in SpA SF can inhibit neutrophil activation and identified hyaluronic acid (HA) as a potential factor preventing excessive neutrophil activation. In the current study, we provide more insight into the behavior of neutrophils in inflamed joints of SpA patients. Furthermore, our data provide arguments for using intra-articular HA supplementation as (additional) therapy in patients with chronic inflammatory joint diseases in order to ensure the inactivity of neutrophils.

## 2. Results

### 2.1. Patient Characteristics

SF samples were collected from 27 patients with an actively inflamed knee joint, of whom 20 were classified as SpA, 5 as RA, and 2 as OA. SF was processed and analyzed immediately after collection. The majority of patients were male (18/27). Age was highly variable (31–85 years), as was the type of treatment, ranging from none to one or more (combinations of) NSAIDs, csDMARDs, and/or bDMARDs. SF derived from these patients contained highly variable amounts of cells (0.2–47.2 × 10^6^ cells/mL) and percentages of neutrophils (2.2–96.4%). The demographics of individuals are listed in Table 1 and Appendix A.

### 2.2. SF-Derived Neutrophils of Severely Inflamed Joints Are Inactive

To determine the activation state of SF-derived neutrophils of SpA patients, we analyzed the neutrophil degranulation status. Neutrophils contain four different types of granules: azurophilic granules, specific granules, gelatinase granules, and secretory vesicles [21,22]. We here assessed neutrophil degranulation by determining the membrane expression of CD16 (FCγRIII); expressed on resting neutrophils and cleaved from the surface by ADAM17 (present in secretory vesicles), CD63 (present in azurophilic granules), and CD66b (present in both specific and gelatinase granules) by flow cytometry [21,22,23,24,25]. Thus, activated neutrophils display an upregulated expression of CD63 and CD66b while CD16 is downregulated compared to unstimulated neutrophils.

We compared the activation status of SF neutrophils from SpA patients (n = 20) with that of blood-derived neutrophils from healthy donors (n = 25), which were either resting or fully activated with GM-CSF (50 U/mL) and LPS (10 ng/mL) [23]. Surprisingly, we found that neutrophils isolated from SF greatly resembled unstimulated blood-derived neutrophils (Figure 1A). Fully activated neutrophils, as defined by CD63high and CD16low, were only observed after GM-CSF+LPS stimulation of blood-derived neutrophils but not of directly measured SpA SF-derived neutrophils (Figure 1A,B). Furthermore, when analyzing single CD16 and CD63 expression, SF-derived neutrophils showed more similarity to unstimulated neutrophils than stimulated neutrophils. Furthermore, SF-derived neutrophils had a low expression of CD66b, which was comparable to unstimulated blood-derived neutrophils. This phenomenon was observed in all 20 SpA SF samples. These samples were derived from SpA patients with high variability in age, gender, medication use, and amount of neutrophils in SF (Table 1), suggesting this phenomenon is independent of these factors. However, to draw conclusions for any relation to age, gender, medication use, and amount of neutrophils, a larger sample set would be necessary.

To determine whether the observed activation status of neutrophils is disease-specific, we also determined the degranulation status of SF-derived neutrophils from RA patients (n = 5) and OA (n = 2). Similar to neutrophils derived from SF from SpA patients, we observed that neutrophils derived from RA and OA SF greatly resembled unstimulated blood-derived neutrophils as well (Appendix A). Although the number of included RA and OA SF samples was much smaller, these findings suggest that inactive neutrophils are present in SF of inflamed joints, which may be irrespective of the disease. Taken together, neutrophils found in SF of patients with SpA greatly resemble unstimulated blood neutrophils and have an inactive phenotype. This observation seemed independent of diagnosis, age, gender, medication, or neutrophil count.

### 2.3. SF Neutrophils from SpA Patients Are Not Exhausted and Can Be Activated Outside of the SF Environment

The observation that SF neutrophils from SpA patients display a non-activated phenotype could be due to the fact that these cells were either not activated yet or were exhausted due to the previous activation. To investigate this, neutrophils were isolated from SF of SpA patients and stimulated in vitro with GM-CSF and LPS. The stimulation of SF neutrophils induced the expression of CD63 and shedding of CD16 (Figure 2A,B). When analyzing CD16 and CD63 expression individually, we also observed a significant decrease in CD16 membrane expression and a significant increase in CD63 membrane expression. Furthermore, we observed a significant increase in CD66b membrane expression compared to unstimulated neutrophils (Figure 2B). In addition to degranulation, another effector function of activated neutrophils is the release of ROS. In line with the observed increase in degranulation, stimulation of SF neutrophils resulted in significantly increased ROS production compared to unstimulated SF neutrophils (Figure 2C,D). For both degranulation as ROS production, no correlation was found between sample characteristics (e.g., neutrophil count) and percentage of degranulation or ROS production. Together, these data demonstrate that neutrophils in SF have an inactive phenotype but are not exhausted as they can degranulate and produce ROS when activated in the absence of SF.

### 2.4. Blood Neutrophil Activation Can Be Inhibited by SF from SpA Patients

To investigate whether certain factor(s) in SF inhibit neutrophil activation, we stimulated blood-derived neutrophils of healthy donors in the presence or absence of SF of SpA patients (30% *v/v*). The presence of SF strongly inhibited the activation of blood-derived neutrophils (Figure 3A,B). This was observed both for degranulation, as demonstrated by the expression of CD63 and CD16 (Figure 3A), and for ROS production (Figure 3B). No correlation was found between sample characteristics (e.g., neutrophil count) and percentage of degranulation or ROS production. Furthermore, titration of SF demonstrated a dose-dependent effect of SF as 30% SF inhibited neutrophil degranulation and ROS production significantly, whereas for lower concentrations (1–3%), no significant effects were observed. (Appendix A). Of note, unstimulated neutrophils were hardly affected by the incubation of SF (Appendix A). Collectively, these data indicate that SF from SpA patients may contain one or more factors that are limiting neutrophil activation.

### 2.5. Hyaluronic Acid in SF of SpA Patients Inhibits Neutrophil Activation

Our results indicate that SF from SpA patients contains factors that restrict neutrophil activation. One of the major components of SF is hyaluronic acid (HA) which is present at a concentration of approximately 1.5–3.1 mg/mL [26]. It was reported that HA can inhibit inflammation in the adjuvant arthritis model [27] and can block neutrophil infiltration and acute lung injury [28], making HA a likely candidate that prevents neutrophil activation in SF. First, we analyzed the expression of CD44, which is a common receptor for HA [29]. SF-derived and blood-derived neutrophils both express CD44 (Figure 4A,B). Next, we determined whether HA at a similar length and concentration range as present in SF of SpA patients was able to inhibit neutrophil degranulation. Indeed, HA at the concentration of 1.6 mg/mL and 3.2 mg/mL was able to significantly inhibit neutrophil degranulation (Figure 4C). Hyaluronidase (hyase) is an enzyme that catalyzes the degradation of HA [30]. We next analyzed whether incubation of SF with hyase could revert the inhibiting effect of HA on neutrophil activation. Hyase alone did not interfere with neutrophil degranulation or ROS production (Appendix A). However, treatment of SF with hyase resulted in a significant but not complete loss of the ability of SF to inhibit neutrophil degranulation (CD63+/CD16−) and ROS production (Figure 4D,E). Taken together our data shows that HA in SF has the capacity to prevent neutrophil activation.

## 3. Discussion

In this study, we show that SF-derived neutrophils of SpA patients are in an inactive state, based on low levels of degranulation and lack of ROS production ex vivo. Similar observations were made for SF-derived neutrophils of RA and OA patients, albeit with a limited sample size. Of note, SF-derived neutrophils are not exhausted as these neutrophils retain their capacity to be activated, degranulate, and produce ROS outside of the SF environment. Moreover, activation of blood-derived neutrophils from healthy donors was dose-dependently inhibited by SF of SpA patients. Further analysis indicated that HA, which is abundantly present in SF, largely contributes to this impediment of neutrophil activation.

It was rather surprising that SF-derived neutrophils are in an inactive state since SF of patients with arthritis contains many neutrophil-activating factors, including TNF and GM-CSF. TNF is present in SF of both treated and untreated SpA or RA patients at a concentration between 94.2 and 378.2 pg/mL or 139.4 and 533.0 pg/mL, respectively [31]. In an earlier study, even higher concentrations of TNF were found in SF of different types of arthritis with average TNF levels of 0.97 (±0.6) ng/mL [32]. GM-CSF is present in SF of patients with various forms of arthritis, including RA and SpA, at concentrations ranging from 5.31 (±3.9) pg/mL to 29.5 (±10.9) pg/mL [33]. Furthermore, the neutrophil-activating factors IL-1β, IFNγ, and TGF-β were found abundantly in SF [19,34]. Taken together, these studies indicate that multiple activating stimuli are present in SF of inflamed joints, which would normally lead to neutrophil activation, especially since dual stimulation of neutrophils with GM-CSF and TNF is efficient in activating neutrophils at relatively low concentrations [23].

Therefore, it is clear that one or more dominant factors are present in SF that prevent neutrophil activation. Here, we show that HA can act as a strong neutrophil inhibiting factor and that hyaluronidase treatment of SF, resulting in the degradation of HA, results in loss of neutrophil inhibitory capacity. HA is known to increase the density of SF, creating a viscous, jelly-like consistency that acts as a lubricant to reduce friction between articular cartilages [35]. To our knowledge, we are the first ones to show that HA in SF of SpA patients prevents different aspects of neutrophil activation. The inhibitory effect of HA on neutrophil activation is not entirely new, but HA can also have pro-inflammatory characteristics [36]. HA has a high molecular mass (HMM) form and a low molecular mass (LMM) form. In SF HMM HA is present in high concentrations (1.5–3.1 mg/mL) [26]. SF derived from arthritic joints contains lower concentrations of HA and reduced chain length compared to SF from healthy joints [37,38]. HA in its HMM form was shown to have immunosuppressive effects [27,28]. In addition, previous studies have demonstrated a clear inhibitory effect of HA on neutrophil ROS production [39] and neutrophil-mediated cartilage degradation [40]. This is in line with our results, showing that HMM HA in high concentrations inhibits neutrophil degranulation and ROS production.

To our knowledge, the effect of SF from SpA patients on neutrophils has not been described before. Although the sample size of other forms of arthritis in the current study is rather small, the data from SF of RA and OA patients suggest that the inhibition of neutrophil activation is a common effect and not only observed in SpA. Other studies have also investigated the effect of SF on neutrophil activation. Two recent studies showed that neutrophils in SF of patients with juvenile idiopathic arthritis (JIA) display an active phenotype based on increased levels of various activation markers, including CD16 and CD66b [41,42]. The same study also tested the effect of 20% SF of JIA patients on healthy blood neutrophils and found no change in degranulation surface markers, including CD16 and CD66b [42]. In other studies that used 10–25% SF of RA patients, induction of NETosis [18,43] and ROS production [18,43,44,45] was found in healthy blood neutrophils, while non-RA (i.e., OA, PsA, and gout) SF inhibited ROS production [45]. Furthermore, previous studies found conflicting results on the effect of SF on apoptosis. While one study has found evidence for enhanced apoptosis of healthy blood-derived neutrophils after the addition of more than 50% SF from RA, SpA, and OA patients after 24 and 48 h [46], another study has found evidence for inhibited apoptosis of healthy blood-derived neutrophils after the addition of 50% SF from RA patients after 12 and 18 h [47]. In our study, we did not observe enhanced or inhibited neutrophil apoptosis. We used different time points and concentrations, looked at neutrophils after 1 and 2 h, and used a maximum of 30%. Consequently, our results might have been different if we looked at other time points or used different concentrations of SF. However, our study was performed with SF from patients with differences in diagnosis, treatment, and potentially, also disease severity, which may, to some extent, account for the somewhat discrepant findings.

In the current study, we showed that HA in SF derived from inflamed joints of SpA patients prevents neutrophil activation. It is known that inflamed joints contain less HA compared to uninflamed joints. Moreover, the capacity of HA to inhibit neutrophil activation decreases significantly when the HA concentration is lower than 1.6 mg/mL. HA supplementation, also known as viscosupplementation, is a therapy that is commonly used in OA patients and is demonstrated to restore lubrication in joints and stimulate the growth of cartilage and bone tissue [48,49]. In our view, patients with other forms of arthritis may also benefit from HA supplementation because, in addition to improving lubrication, we demonstrate that it can also prevent (neutrophil-induced) inflammation. Importantly, the effects of intra-articular (IA)-HA injection were described to be much longer-lasting than IA corticosteroid injection (approximately 6 months vs. 1–2 months, respectively) and have no significant adverse effects [48]. However, IA-HA injections are not commonly used in arthritis types other than OA. Of note, one study showed that IA-HA injection is beneficial in RA patients [50]. In the current study, we demonstrated that using HMM-HA inhibits neutrophil activation. Therefore, we propose that HA supplementation may hold great potential in some arthritis patients and have beneficial effects that reach beyond improving the viscosity of SF and enhancing lubrication, since it might also reduce inflammation. This may be especially of value for patients with chronic monoarthritis that is persistent even with adequate systemic treatment. This may ultimately result in dose reduction or decrease the number of DMARDs that patients require and improve quality of life.

In conclusion, the present study demonstrates that SF-derived neutrophils of SpA patients display an inactive phenotype, although they are not exhausted as these neutrophils can be activated to degranulate and produce ROS outside the SF micro-environment. Importantly, we demonstrate that SF can also inhibit the activation of blood-derived neutrophils from healthy donors. Finally, we show that HA present in SF can act as a strong inhibitor of neutrophil activation and that hyaluronidase treatment of SF, resulting in the degradation of HA, results in the loss of neutrophil inhibitory capacity. Our study can, therefore, be considered as an argument to look at HA as a potential novel treatment option for chronic inflammatory joint diseases, although this remains to be formally tested in a randomized controlled trial.

## 4. Materials and Methods

### 4.1. Synovial Fluid Collection and Preparation of Synovial Cells and Cell-Free Synovial Fluid

Synovial fluid (SF) from inflamed knee joints was collected during active arthritis from 27 patients after obtaining informed consent. Patients’ characteristics are described in Table 1 and Appendix A. SF was centrifuged at 650× *g* for 20 min to pellet the cells. The SF was centrifuged at 3000× *g* for 30 min at RT to pellet all remaining cells and debris; the cell-free SF was collected and stored at −80 °C until further use. Meanwhile, the SF-derived cells collected after the first 650 g step were resuspended in IMDM (Gibco; Thermo Fischer Scientific Inc., Waltham, MA, USA) supplemented with 10% heat-inactivated (HI) fetal bovine serum (FBS; Hyclone; Thermo Fischer Scientific Inc., Waltham, Mass) and gentamycin (86 µg/mL; Duchefa Biochemie B.V., Haarlem, The Netherlands) and passed through a 70 µm single-cell filter. Then, cells were resuspended in IMDM with 10% HI FBS at a concentration of 2 × 10^6^ cells/mL for flow cytometry analysis and culture experiments.

### 4.2. Neutrophil Isolation from Blood

Blood was collected from healthy volunteer donors after obtaining informed consent into sodium heparin tubes (Greiner Bio-One, Alphen a/d Rijn, The Netherlands). Neutrophils were isolated using density gradient followed by erythrocyte lysis, as previously described [23]. Neutrophils were then resuspended in IMDM (Gibco; Thermo Fischer Scientific Inc., Waltham, MA, USA) supplemented with 10% heat-inactivated (HI) fetal bovine serum (FBS; Hyclone; Thermo Fischer Scientific Inc., Waltham, MA, USA) and gentamycin (86 µg/mL; Duchefa Biochemie B.V., Haarlem, The Netherlands) and used immediately. Neutrophil purity was analyzed by flow cytometry and was always >97%.

### 4.3. Neutrophil Culture

Neutrophils were seeded at a density of 0.4 × 10^6^ cells/mL in 250 µL in a flat-bottom 96-well plate (Costar, Corning Inc., Corning, NY, USA) in an IMDM medium containing 10% HI-FBS and gentamycin. Subsequently, neutrophils were cultured for 1 or 2 h at 37 °C (CO_2_ incubator) in the absence or presence of granulocyte-macrophage colony-stimulating factor (GM-CSF) (Schering-Plough B.V., Brussels, Belgium) and lipopolysaccharide (LPS) (Sigma-Aldrich) and the absence or presence of SF, hyaluronic acid (HA, Sigma-Aldrich, St Louis, MO, USA), and SF pretreated with hyaluronidase (hyase, Sigma-Aldrich; 40 U/mL). For flow cytometric analysis of ROS production, neutrophils were cultured and stimulated for 1 h in the presence of 25 µM 123-dihydrorhodamine (123-DHR; Marker Gene Technologies, Eugene, OR, USA). For the flow cytometric analysis of CD63, CD66b, and CD16, neutrophils were stimulated for 2 h. After stimulation, cells were harvested and used for measurement of degranulation or ROS production.

### 4.4. Measurement of Degranulation Markers

Cells were washed twice at 4 °C in PBA (PBS-0.5% *w/v* BSA-0.05% *w/v* azide), followed by antibody labeling in PBA. CD16, CD63, and CD66b expression and cell viability (PI) were determined using flow cytometric analysis. The following antibodies were used: αCD15-FITC (1:100; HI98), αCD16-PECy7 (1:1000; 3G8), αCD63-APC (1:100; H5C6), αCD66b-PE (1:100; G10F5), (all Biolegend, San Diego, CA, USA). Propidium iodide (PI) (Sigma-Aldrich) was used to determine cell viability. A total of 10,000 cells were acquired in the live gate on a FACSCanto (BD Biosciences, San Jose, CA, USA) and further analyzed using FlowJo software version 10.7.1 (BD Biosciences).

### 4.5. Measurement of ROS Production

Cells were washed twice at 4 °C in PBA (PBS-0.5% *w/v* BSA-0.05% *w/v* azide) and measured using flow cytometry. A total of 10,000 cells were acquired in the live gate on a FACSCanto (BD Biosciences) and further analyzed using FlowJo software (BD Biosciences).

### 4.6. Statistical Analysis

Data are expressed as mean ± SD or as mean and individual points. Statistical analysis was performed in Graphpad Prism version 9.1.0 for Windows by using statistical tests, depending on experimental data. The Shapiro–Wilk test was performed to test the normality of data. For single comparisons, *p* values were calculated using two-tailed paired *t*-tests. For multiple comparisons, *p*-values were calculated using a one-way ANOVA. *p*-values below 0.05 were considered statistically significant.

## Figures and Tables

**Figure 1 ijms-24-03066-f001:**
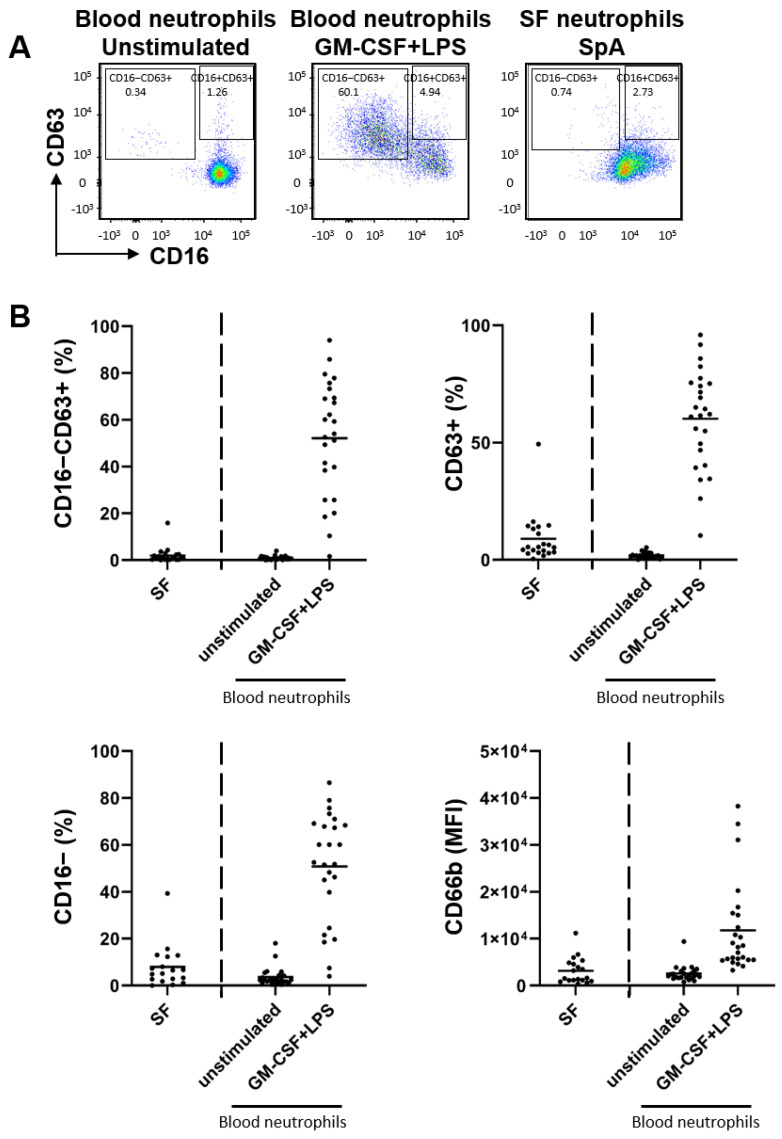
Neutrophils derived from SF of SpA patients do not degranulate. Neutrophils from SF of SpA patients (n = 20) were analyzed by flow cytometry and compared to neutrophils derived from blood of healthy donors (n = 25). (**A**) Representative flow cytometry plot of unstimulated healthy donor-derived blood neutrophils, stimulated blood neutrophils, and SF-derived neutrophils of a SpA patient demonstrating CD16 and CD63 membrane expression. (**B**) Full neutrophil degranulation as measured by percentage of CD16−CD63+ neutrophils, secretory vesicle degranulation as measured by percentage of CD16− neutrophils, azurophillic degranulation as measured by percentage of CD63+ neutrophils, and specific and gelatinase degranulation as measured as mean fluorescent intensity (MFI) of CD66b. Data are presented as mean and individual points.

**Figure 2 ijms-24-03066-f002:**
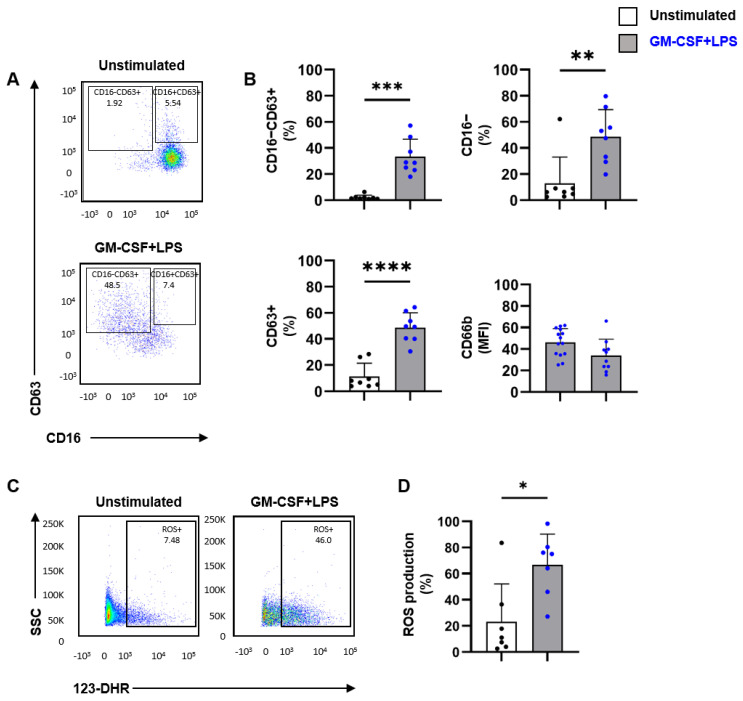
Neutrophils derived from SF of SpA patients are able to degranulate and produce ROS. (**A**) SF cells of SpA patients (n = 8, pat# 4, 5, 7, 8, 10, 11, 12, 15) were cultured for 2 h in the absence or presence of GM-CSF (50 U/mL) and LPS (10 ng/mL), Representative flow cytometry plot of synovial fluid neutrophils demonstrating CD16 and CD63 membrane expression for unstimulated and GM-CSF+LPS stimulated cells. (**B**) Percentage of CD16−CD63+, CD16−, CD63+, and MFI of CD66b membrane expression as a measure for degranulating neutrophils. (**C**) SF cells of SpA patients (n = 7, pat# 10, 11, 12, 15, 17, 19, 20) were incubated for 1 h in the presence of 123-DHR. Representative flow cytometry plot of SF neutrophils demonstrating ROS production for unstimulated and GM-CSF+LPS stimulated cells. (**D**) Intracellular ROS generation expressed as percentage of ROS+ cells. Data are presented as mean ± SD. * *p* < 0.05, ** *p* < 0.01, *** *p* < 0.001, and **** *p* < 0.0001. The *p*-values were calculated using a paired *t*-test.

**Figure 3 ijms-24-03066-f003:**
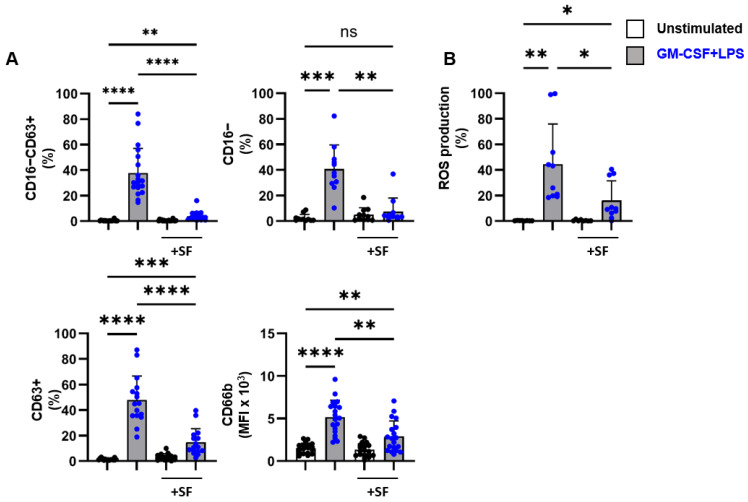
SF of SpA patients inhibits activation of healthy blood-derived neutrophils. (**A**) Blood-derived neutrophils from healthy donors (n = 11–15) were cultured for 2 h in the absence or presence of GM-CSF (50 U/mL) and LPS (10 ng/mL) and with or without 30% SF of SpA patients (n = 9–12, pat# 7, 8, 10, 12–20). Percentage of CD16-CD63+, CD16-, CD63+, and MFI of CD66b membrane expression as a measure for degranulating neutrophils. (**B**) Blood-derived neutrophils from healthy donors (n = 8) were cultured for 1 h in the presence of 123-DHR and GM- CSF (50 U/mL) and LPS (10 ng/mL) and with or without 30% SF of SpA patients (n = 6, pat# 13, 14, 16–19). Percentage of ROS production. Data are presented as mean ± SD. * *p* < 0.05, ** *p* < 0.01, *** *p* < 0.001, and **** *p* < 0.0001. The *p*-values were calculated using a one-way ANOVA.

**Figure 4 ijms-24-03066-f004:**
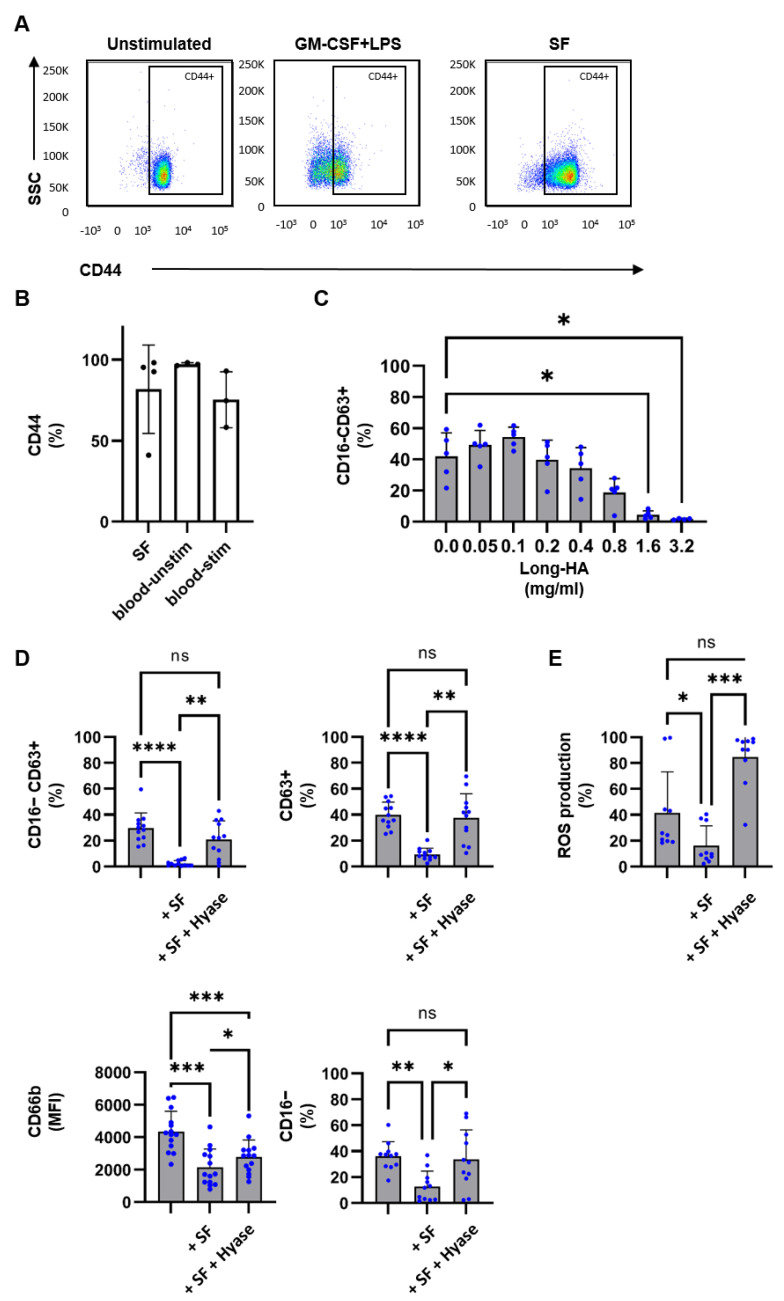
HA in SF inhibits neutrophil activation. (**A**,**B**) Blood-derived and SF-derived neutrophils were cultured for 2 h in the absence or presence of GM-CSF (50 U/mL) and LPS (10 ng/mL). (**A**) Representative flow cytometry plot of unstimulated blood neutrophils, stimulated blood neutrophils, and synovial fluid neutrophils demonstrating CD44 membrane expression. (**B**) Percentage of CD44 membrane expression (n = 3–4, pat# 16, 18, 19, 20). (**C**) Blood-derived neutrophils were cultured for 2 h in the presence of GM-CSF (50 U/mL) and LPS (10 ng/mL) and with various concentrations of HA. Percentage of CD16-CD63+ as a measure of full degranulation shown. (**D**) Blood-derived neutrophils from healthy donors (n = 11–19) were cultured for 2 h in the presence of GM-CSF (50 U/mL) and LPS (10 ng/mL), without SF or with 30% SF or with 30% SF treated with hyase. SF was obtained from SpA patients (n = 9–12, pat# 7, 8, 10, 12–20). Percentage of CD16-CD63+, CD16-, CD63+, and MFI of CD66b membrane expression as a measure for degranulating neutrophils. (**E**) Blood-derived neutrophils from healthy donors (n = 10) were cultured for 1 h in the presence of 123-DHR and GM- CSF (50 U/mL) and LPS (10 ng/mL), without SF or with 30% SF of with 30% SF treated with hyase. SF was obtained from SpA patients (n = 6, pat# 13, 14, 16–19). Percentage of ROS production. Data are presented as mean ± SD. * *p* < 0.05, ** *p* < 0.01, *** *p* < 0.001, and **** *p* < 0.0001. The *p*-values were calculated using a one-way ANOVA.

**Table 1 ijms-24-03066-t001:** Description of the patient cohort and description of SF data from SpA patients. ND = not determined.

Patient#	Diagnosis	Gender M/F	Age Years	Treatment	Type of bDMARd (ia)	Total Cells in SF (cells/mL)	Amount of Neutrophils in SF (%)
1	SpA	F	50	NSAID		ND	ND
2	SpA	M	59	NSAID, bDMARD	anti-IL17	1.6 × 10^6^	77.9
3	SpA	M	34	NSAID, csDMARD		20.2 × 10^6^	74.7
4	SpA	M	52	bDMARD	anti-TNF	4.5 × 10^6^	82.3
5	SpA	M	31	None		7.8 × 10^6^	42.1
6	SpA	F	58	bDMARD	anti-TNF	ND	25.1
7	SpA	M	75	NSAID, bDMARD	anti-TNF	ND	86.9
8	SpA	M	35	NSAID		ND	27.0
9	SpA	M	75	NSAID		10.1 × 10^6^	96.4
10	SpA	F	48	NSAID, bDMARD	anti-IL17	1.5 × 10^6^	2.2
11	SpA	M	35	NSAID, csDMARD, bDMARD	anti-TNF	47.2 × 10^6^	39.4
12	SpA	M	55	bDMARD	anti-TNF	8.0 × 10^6^	48.7
13	SpA	F	58	csDMARD, bDMARD	anti-TNF	2.6 × 10^6^	44.4
14	SpA	M	36	NSAID, csDMARD, bDMARD	anti-TNF	38.1 × 10^6^	66.8
15	SpA	F	58	NSAID, csDMARD, bDMARD	anti-IL17	2.8 × 10^6^	16.1
16	SpA	M	36	NSAID, csDMARD, bDMARD	anti-TNF	22.3 × 10^6^	33.6
17	SpA	M	64	csDMARD		0.2 × 10^6^	40.9
18	SpA	F	52	csDMARD, bDMARD	anti-TNF	4.5 × 10^6^	8.0
19	SpA	M	56	csDMARD		4.4 × 10^6^	80.9
20	SpA	F	36	NSAID		6.6 × 10^6^	62.8

## Data Availability

The data presented in this study are available on request from the corresponding author.

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
