# Peer review of "Hyaluronic Acid in Synovial Fluid Prevents Neutrophil Activation in Spondyloarthritis"

_ijms, 2023, doi:10.3390/ijms24043066_

Round 1
Reviewer 1 Report
Review of Hyaluronic acid in synovial fluid prevents neutrophil activation in spondyloarthritis by Mol et al.
Major comments;
- The main issue is of this research is to explain how inactive neutrophils relate to inflammation? In other words, what does it mean that in an active inflamed joint neutrophils do not seem to have a role. Is it other cells that contribute to the inflammation? These results are not in line with other papers that show that intraarticular injection of hyaluronic acid relieves clinical symptoms in osteoarthritis patients. The authors do not comment upon this subject.
- SF was probably obtained from inflamed joints. The authors do not mention how much fluid was obtained and also not CRP levels and other information on clinical factors.
- Is there a relation with therapy? IL-17 is produced by neutrophils and anti-IL-17 therapy is given to some of the patients.
- There is lots of variation in neutrophil counts in SF from patients. What is known on lymphocyte en macrophage count in the SF, since these cells are also present?
- Is the level of hyaluronic acid known in SF? Is there a large variation too? It is necessary to know this in order to determine whether intra-articular injection of hyaluronic acid is needed or not.
- The authors do not give information on which samples of SF were used for the invitro experiments. Was this with high or low neutrophil counts. This might be important.
Minor comments:
- The text in the FACS plots is hard to read
Reviewer 2 Report
1) The microenvironment of rheumatoid arthritis SF, is considered as a proinflammatory milieu. Contrary to what has been previously identified, https://www.ncbi.nlm.nih.gov/pmc/articles/PMC7813679/
https://www.ncbi.nlm.nih.gov/pmc/articles/PMC4229860/ , author found that neutrophils from SF of patients with SpA, RA, and OA joint inflammation suppressed neutrophil activation and function. Author needs to provide additional evidences and mechanistic insights to support this hypothesis.
2) Author only provided flow cytometry-based data to make claims about neutrophil activation and function in presence of SF. Author needs to validate these findings with additional functional tests such as Cell migration by Transwell assay, NETs formation assay by fluorescence microscopy, fluorescence gelatin degradation assay, ELISA of MPO, Elastase etc.
3) Only 1 activator LPS, was used. Additional activators of neutrophil such as PMA, FMLP, TNF-a, Ca-I should be tested.
4) 2 OA and 5RA sample number is too low to draw any conclusion.
5) Supplemental figure 2- What was the positive control for ROS and CD16 measurements? X-axis of the graph should be reformatted to reflect changes between groups.
6) Missing description about Supp Fig 2C, 2D in the legend.
Round 2
Reviewer 1 Report
There are no further comments.
Author Response
Dear reviewer,
We thank you for reviewing our manuscript. We have carefully checked the manuscript for typo's and spelling mistakes and have adjusted these.
Kind regards, Esther de Jong
Reviewer 2 Report
Why there is no stats on figure 1B and supplemental figure 1?
Author Response
Dear reviewer,
We thank you for carefully reviewing our manuscript. We have carefully checked the manuscript for typo's and spelling mistakes and have adjusted these.
In addition, you questing why Figure 1B and Suppl. Figure 1 do not have any statistical analysis.
The data depicted in Figure 1 and Suppl. Figure 1 show that SF-derived neutrophils have not a degranulated fenotype. In comparison we show blood neutrophils degranulation in a stimulated condition. Therefore we do not warrant these data suitable for statistical analysis. However, if the reviewer or the editor insists, we will perform the statistical analysis and add it to the manuscript.
Kind regards, Esther de Jong